Methods

# Targeted variant detection using unaligned RNA-Seq reads

Eric Olivier Audemard[1] , Patrick Gendron[1] , Albert Feghaly[1] , Vincent-Philippe Lavallée[1,2], Josée Hébert[1,2,4,5], Guy Sauvageau[1,2,4,5], Sébastien Lemieux[1,3]

**Mutations identified in acute myeloid leukemia patients are useful for prognosis and for selecting targeted therapies. Detection of such mutations using next-generation sequencing data requires a computationally intensive read mapping step followed by several variant calling methods. Targeted mutation identification drastically shifts the usual tradeoff between accuracy and performance by concentrating all computations over a small portion of sequence space. Here, we present *km*, an efficient approach leveraging k-mer decomposition of reads to identify targeted mutations. Our approach is versatile, as it can detect single-base mutations, several types of insertions and deletions, as well as fusions. We used two independent cohorts (The Cancer Genome Atlas and Leucegene) to show that mutation detection by *km* is fast, accurate, and mainly limited by sequencing depth. Therefore, *km* allows the establishment of fast diagnostics from next-generation sequencing data and could be suitable for clinical applications.**

## Introduction

Extensive molecular characterization of patient samples forms the basis of precision medicine and has proven useful in a number of pathologies, including acute myeloid leukemia (AML) (Prada-Arismendy et al, 2017). The use of RNA-Seq for sample characterization is appealing because (i) it is unbiased, as library preparation does not require capture or amplification, (ii) it focuses on a very small portion of the genome (~2%), namely, the transcribed one, and (iii) mutations in regulatory regions are expected to leave a trace on the transcriptome, such as altering transcript expression. Unfortunately, the diversity of splicing events as well as incomplete transcriptome annotations make the mapping of RNA-Seq reads difficult and computationally intensive and may introduce mapping errors, which are all inherited by variant calling methods.

In both clinical and research settings, analysis is often focused on a very limited set of expected variants (e.g., specific genes, positions, fusions, or splice variants) for which some predictive value has been established. For instance, the FLT3 internal tandem duplication (FLT3–ITD) represents such a case for AML because its presence is associated with poor prognosis (Fröhling et al, 2002; Dohner et al, 2017). Therefore, clinicians routinely assess the presence of FLT3–ITD in patients using PCR-based tests (Gale et al, 2008; Stone et al, 2017) on a targeted region. Alternatively, attempts to detect FLT3–ITD in next-generation sequencing (NGS) data have been hampered by the difficulties of read mappers to properly align reads overlapping repeat junctions. To circumvent this problem, recent tools such as ITDseek (Au et al, 2016) and ITDassembler (Rustagi et al, 2016) detect ITD from soft-clipped reads and require BAM files specifically generated by BWA-MEM (Li, 2013 *Preprint*), which does not perform an end-to-end alignment. Similarly, the best tools highlighted by Liu et al (2016) to detect gene fusions combine multiple read alignment methods to achieve higher accuracy of both read alignment and fusion breakpoint detection. These methods spend a considerable amount of resources mapping reads outside areas of interest. They might also lose reads because of incorrect mappings. This is especially damaging if the incorrect mapping is due to the presence of a variant. This situation is most acute for variants resulting from sequence rearrangements (e.g., duplications, inversions, and fusions), prompting the development of specialized software tools to detect them from NGS data.

Here, we present *km* (https://github.com/iric-soft/km), a method for targeted variant detection that shares algorithmic similarity with local assembly variant calling methods, such as SvABA (Wala et al, 2018), but without the requirement for mapping reads to a reference transcriptome (or genome). This idea is shared with the author of TASR (Warren & Holt, 2011), which performs an assembly of targeted unaligned reads. Unfortunately, selecting reads hosting a targeted variant while preserving the locality and relevance of the assembly is a difficult task. Instead, *km* looks for evidence of mutations using a compact digest of unaligned reads, similarly to LAVA (Shajii et al, 2016) but without being limited to single

[1]The Leucegene Project at Institute for Research in Immunology and Cancer, Université de Montréal, Montréal, Canada    [2]Division of Hematology, Maisonneuve-Rosemont Hospital, Montréal, Canada    [3]Department of Biochemistry, Université de Montréal, Montréal, Canada    [4]Quebec Leukemia Cell Bank, Maisonneuve-Rosemont Hospital, Montréal, Canada    [5]Department of Medicine, Faculty of Medicine, Université de Montréal, Montréal, Canada

Correspondence: s.lemieux@umontreal.ca

nucleotide variants (SNVs). We demonstrate the usefulness of our method by detecting a set of several types of variants, which contribute to the identification of AML prognostic subgroups, on two large RNA-Seq cohorts. The Leucegene cohort comprises 437 deep transcriptomes (average number of reads: $204 \times 10^{6}$) from AML patient samples. The second cohort, consisting of 10,407 samples from 33 cancer types (including 151 AML) from TCGA, is analyzed mostly as a technical demonstration of *km*'s efficiency. Finally, we compare our in silico results with the experimental results of two clinical laboratories.

# Results

### Overview of *km* and target sequences

Taking advantage of its targeted nature, *km* performs an extensive analysis of a single user-defined sequence, called the *target sequence*, which contains the region of interest. This *target sequence* is broken down into k-mers (subsequences of length k) overlapping by k-1 bp, with *k* large enough to produce a linear directed graph (Fig 1A). Independently, a k-mer count table is prepared, reporting the occurrence of each k-mer in the sequenced sample (Marcais & Kingsford, 2011). A sequenced variant will then appear as an alternative path connecting the starting and ending k-mers, detected by walking along the linear directed graph and following new overlapping k-mers queried from the count table. Similarly to a colored de Brujin graph, used by Cortex (Iqbal et al, 2012) and others (Gnerre et al, 2011), this process returns a graph as an approximate local assembly, which is then used to identify the presence of simple (e.g., single-base mutation) or more complex sequence variants, for example, indels and other insertions or deletions (Fig 1B). In the absence of an alternative path, the target sequence has no variants in sequenced reads.

To interpret km's output, which is directly dependent on the sequence given as input, we defined two types of target sequences. *Reference target* represents all sequences extracted directly from the reference genome or transcriptome. This type of sequence is ideal for finding small variant such as SNVs (Fig 1C), insertions, deletions, indels, and small ITDs (Fig 1D). For structural variants, a *variant target* is generally more suitable. Sequences of this type are designed to represent the expected mutation, in which case an alternative path to the *target sequence* indicates either an alternative mutation or the reference (Fig 1E).

**Figure 1. Overview of km.**
**(A)** Visual description of how km detects the DNMT3A R882 SNP, using k-mers of 31 bp. The input sequence (target sequence given by user) is centered on DNMT3A's 882nd codon. This sequence is segmented in k-mers to create a linear graph, which represents the search space delimited by the starting and ending k-mers (hatched). **(B)** A variant will be represented by a new path between the two extremities. This path is found by walking along the linear directed graph and following new (i.e., not seen in the target sequence) overlapping k-mers, queried from a sample's count table. **(C)** Schematic representation of an SNP k-mer graph with each path representing the target (lower) and the variant (upper) sequences. **(D)** Same representation for an ITD variant. **(E)** Same representation showing the use of a *variant target sequence* to initiate km's search. Here, the expected variant is detected when all k-mers that overlap the starting and ending k-mers (in red) are found in the sample's count table. Also, additional variants could be detected if another path is found (in orange).

Using these two types of sequences, Table 1 reports our current catalog, which was designed to represent several types of mutations and illustrate strategies used by *km* to detect them. It includes SNVs in IDH1 (Medeiros et al, 2017), DNMT3A (Yuan et al, 2016), and MYC (Lavallée et al, 2016); insertions in NPM1 (Thiede et al, 2006) and ITD (Shiba et al, 2016); and mutations in the tyrosine kinase domain of FLT3 (Bacher et al, 2008); partial tandem duplication (PTD) in KMT2A; and a fusion between NUP98 and NSD1 (Lavallée et al, 2016; Shiba et al, 2016). Moreover, these variants have already been validated, clinically or with other in silico methods, in both TCGA and Leucegene cohorts (Network et al, 2013; Lavallée et al, 2015, 2016). To our knowledge, all these variants are specific to AML except the SNV in IDH1 and are also used for prognosis in glioma tumors (Yan et al, 2009) (Table S1).

### Cohort annotation

Applying *km* to the 437 samples from the Leucegene cohort for all *target sequences* in Table 1 required 3 h and 11 min of CPU time (an average of 26 s per sample), which excludes time taken for pre-computing count tables (10 min per sample). Starting from sequenced reads, the Leucegene cohort can thus be annotated for nine variants in less than 4 d, on a single workstation (i7-6700K at 4.00 GHz, using four threads, 8 GB of RAM, and 31-bp-long k-mers). In addition, the required storage space is about four times smaller for the k-mer count tables than aligned reads in BAM format, with an average file size of 3.2 GB per sample (see Fig S1).

Table 2 presents the outcome of *km* on the catalog of *target sequences* for the Leucegene and TCGA cohorts. Columns in "km type" indicate the specific *km* annotations returned for each variant. In particular, *km* reports insertions using four subtypes based on the composition of the inserted sequence. When the insertion is adjacent to a deletion, *km* flags the variant as an indel. When the inserted sequence is identical to a part of the *target sequence*, *km* returns an ITD. An I&I (Insertion and ITD) is returned when the inserted sequence is not identical but has

more than 50% identity with the duplicated part. If none of these conditions describe the inserted sequence, *km* returns an Insertion. The TCGA cohort is divided in two subgroups, 151 AML and 10,256 non-AML samples, to show the specificity of each variant found.

### Sensitivity and precision

We assessed sensitivity and precision of *km* based on the detection of insertions in NPM1 and FLT3 (see Table S2). These two variants have been experimentally validated by the Banque de Cellules Leucémique du Québec (BCLQ, www.bclq.org) and TCGA (Network et al, 2013), independently from RNA-Seq. For FLT3–ITD, we also compared *km* with existing methods previously proposed to detect ITD (e.g., ITDassembler [Rustagi et al, 2016], Pindel [Ye et al, 2009], and Genomon ITDetector [Chiba et al, 2015]). We performed limited tests using TASR (Warren & Holt, 2011) with our NPM1 and FLT3–ITD target sequences. On seven Leucegene and three TCGA AML samples, we find that it returns a large number of mutated sequences (even on nonmutated samples), without providing annotations on the type and location of possible variants. This format of output impedes the application of the method on a scale as a big as the entire Leucegene or TCGA cohorts.

On NPM1 variants, *km* successfully identified 117 mutated Leucegene samples of the 118 clinically validated cases by the BCLQ (on 202 tested). All mutated samples identified by *km* were either validated independently by the BCLQ or not part of the subset of samples tested clinically. This high level of performance is also observed with the AML subset of the TCGA cohort (Network et al, 2013), where *km* identified all clinically validated mutated samples (36 clinically validated of 150 tested), in addition to four mutated samples not detected in clinical settings. Nevertheless, all 4-bp insertions reported by *km* are also known variants in the COSMIC database (see Table 3).

Similar performance is observed for FLT3–ITD, where *km* identified all variants clinically validated from the Leucegene cohort

**Table 1. Current catalog of targeted mutations for AML.**

| Gene | Expected type of variant | Gene location | Target sequence length (bp) | Average running time[a] (s/sample) |
|---|---|---|---|---|
| Reference target | | | | |
| IDH1 | SNP (R132) | Exon 4 | 65 | 0.008 |
| DNMT3A | SNP (R882) | Exon 23 | 65 | 0.01 |
| NPM1 | 4-bp insertion | Exon 10–11 + UTR | 80 | 0.017 |
| FLT3 | ITD | Exon 13–15 | 345 | 0.107 |
| FLT3 | TKD | Exon 20 | 68 | 0.019 |
| MYC | SNP (T58A/P59R) | Exon 2 | 68 | 0.018 |
| Variant target | | | | |
| NUP98–NSD1 | Fusion | Exon 11 + 7 | 62 | 0.07 |
| NSD1–NUP98 | Fusion | Exon 6 + 13 | 62 | 0.061 |
| KMT2A | PTD | Exon 8 + 2 | 62 | 0.067 |

[a]Average computation times are reported for Leucegene samples and assume that k-mer count tables are cached in RAM before running *km.* The performance of the caching step is highly dependent on I/O architecture, taking around 25 s on a typical system. The approaches used to prepare each *target sequence* for detecting the expected mutations are presented in the Materials and Methods section.

**Table 2.  Variants identified by km using our AML catalog in the Leucegene and TCGA cohort.**

| Dataset | Mutation name | Km type | | | | | | Variant | Target | Number of samples |
|---|---|---|---|---|---|---|---|---|---|---|
| | | Ins | Del | Indel | Sub | ITD | I&I | | | |
| Leucegene | IDH1 R132 | 0 | 0 | 0 | 32 | 0 | 0 | **32** | 437 | 437 |
| | DNMT3A R882 | 0 | 0 | 0 | 64 | 0 | 0 | **64** | 436 | |
| | NPM1 4-bp ins | 22 | 0 | 1 | 0 | 103 | 13 | **139** | 437 | |
| | FLT3–ITD | 10 | 3 | 3 | 38 | 83 | 54 | **162** | 429 | |
| | FLT3–TKD | 0 | 4 | 0 | 31 | 0 | 0 | **34** | 434 | |
| | MYC T58A/P59R | 0 | 0 | 0 | 2 | 0 | 0 | **2** | 437 | |
| | *NUP98–NSD1* | *7[a]* | *0* | *0* | *0* | *0* | *0* | ***7*** | ***6*** | |
| | *NSD1–NUP98* | *0* | *0* | *0* | *0* | *0* | *0* | ***0*** | ***2*** | |
| | *KMT2A–PTD* | *10[b]* | *0* | *0* | *0* | *0* | *0* | ***10*** | ***15*** | |
| TCGA (AML) | IDH1 R132 | 0 | 0 | 0 | 11 | 0 | 0 | **11** | 148 | 151 |
| | DNMT3A R882 | 0 | 0 | 0 | 12 | 0 | 0 | **12** | 149 | |
| | NPM1 4-bp ins | 6 | 0 | 0 | 0 | 28 | 6 | **40** | 151 | |
| | FLT3–ITD | 76 | 0 | 5 | 22 | 20 | 18 | **100** | 142 | |
| | FLT3–TKD | 0 | 1 | 0 | 11 | 0 | 0 | **12** | 149 | |
| | MYC T58A/P59R | 0 | 0 | 0 | 2 | 0 | 0 | **2** | 139 | |
| | *NUP98–NSD1* | *0* | *0* | *0* | *0* | *0* | *0* | ***0*** | ***0*** | |
| | *NSD1–NUP98* | *0* | *0* | *0* | *0* | *0* | *0* | ***0*** | ***2*** | |
| | *KMT2A–PTD* | *3[b]* | *0* | *0* | *0* | *0* | *0* | ***3*** | ***0*** | |
| TCGA (non-AML) | IDH1 R132 | 0 | 0 | 0 | 394 | 0 | 0 | **394** | 9,850 | 10,256 |
| | DNMT3A R882 | 0 | 0 | 0 | 0 | 0 | 0 | **0** | 9,267 | |
| | NPM1 4-bp ins | 0 | 0 | 0 | 0 | 0 | 0 | **0** | 1,0232 | |
| | FLT3–ITD | 0 | 0 | 0 | 9 | 0 | 0 | **9** | 163 | |
| | FLT3–TKD | 0 | 0 | 0 | 0 | 0 | 0 | **0** | 361 | |
| | MYC T58A/P59R | 0 | 0 | 0 | 5 | 0 | 0 | **5** | 9,204 | |
| | *NUP98–NSD1* | *0* | *0* | *0* | *0* | *0* | *0* | ***0*** | ***0*** | |
| | *NSD1–NUP98* | *0* | *0* | *0* | *0* | *0* | *0* | ***0*** | ***0*** | |
| | *KMT2A–PTD* | *0* | *0* | *0* | *0* | *0* | *0* | ***0*** | ***0*** | |

[a]Fusion with exon 12 found as an insertion in the target sequence.
[b]Tandem duplication extended with exon 9 or 9 and 10.
Each dataset is split into two parts: reference target and variant target (italic). The "target" column reports the number of samples expressing the target sequence. The "variant" column shows the number of samples where at least one variant of the target sequence is found. As a variant target sequence represents a mutated sequence (Fig 1E), mutated samples counts are indicated in bold. The columns in "km type" identify the specific types of variants detected. Of note, several types of variants can be identified in a given sample. As expected, SNVs on IDH1 are found in AML and non-AML samples on lower grade glioma (LGG) (see Table S1).

and 31 of 33 cases for the TCGA AMLs. Upon further investigation of *km*'s output for these two missing samples (IDs TCGA-AB-2812 and TCGA-AB-2988), we found that the expected ITD sequences are indeed detected, but the variant sequences were filtered out of the final output because of part of the *target sequence* having zero coverage (before or after the diverging path, see the Materials and Methods section). We opted to filter out these sequences by default, as they can arise from the expression of a region made up of k-mers derived from a different transcript composed by subsequences in common with the *target sequence*. However, this is not the case for these two samples where identification of the ITDs is missing because of a homozygous SNV after the ITD (t/C at chr13:28,034,322).

Unfortunately, *km* is designed to independently report single mutation events for a *target sequence* in a given sample. Here, the homozygous SNV is detected by *km* and can be flagged as a potential false negative or used to modify the *target sequence* to find the FLT3–ITD in a second *km* analysis. Other strategies are currently being explored to account for the possibility of concurrent mutations for a given target without compromising speed, sensitivity, and precision.

Until now, specialized methods for detecting ITDs were designed for exome sequencing, such as ITDassembler, Pindel, and Genomon ITDetector. Consequently, comparison with *km* was performed on a reduced TCGA AML cohort of 28 samples shared by all studies (33

**Table 3.  NPM1 mutations identified by km.**

| Type | COSMIC ID | Km variant | Km type | Leucegne | TCGA AML |
|------|-----------|------------|---------|----------|----------|
| A | COSM17559 | TCTG | ITD | 103 | 28 |
| D | COSM17573 | CCTG | I&I | 11 | 4 |
| B | COSM17571 | CATG | Insertion | 10 | 4 |
| | COSM20809 | CCAG | Insertion | 3 | 0 |
| | COSM29814 | CAGA | Insertion | 2 | 0 |
| | COSM3356078 | CAAG | Insertion | 1 | 0 |
| | COSM29814 | CCGA | Insertion | 1 | 0 |
| | COSM3356078 | CGCG | Insertion | 1 | 1 |
| | COSM28066 | CGGA | Insertion | 1 | 0 |
| | COSM20811 | CTTG | Insertion | 1 | 0 |
| | COSM20815 | TATG | I&I | 1 | 2 |
| | COSM20813 | TCGG | I&I | 1 | 0 |
| | COSM27390 | TTCG | Insertion | 1 | 0 |
| | COSM20850 | AGAA | Insertion | 1 | 0 |
| | COSM20810 | TTGT | Insertion | 0 | 1 |
| | Unknown | gc/CAGGG | Indel | 1 | 0 |
| | Total mutated/total | | | 138/437 | 40/151 |

samples reported by ITDassembler [Rustagi et al, 2016], five of which had no RNA-Seq data available). Of these 28 samples, 22 had clinically validated FLT3–ITD annotations (Network et al, 2013). On this reduced cohort, *km* has a sensitivity of 95% (21 of 22), whereas ITDassembler identified 15 samples, Genomon 14 (one had an incorrect duplication length), and Pindel 11 (five had incorrect duplication lengths). As mentioned by ITDassembler's author (Rustagi et al, 2016), five of the seven missed mutated samples can be attributed to the length of the duplication, which approaches or exceeds the read length (>60 bp in a 75-bp read). By contrast, the nature of *km*'s algorithm does not limit the duplication size to the read length but instead relies on a user-defined parameter that limits the maximum length of alternative branches to explore (default 500 bp). All details of ITDs found by each software can be found in Table S3.

### Sensitivity and coverage in RNA-Seq

To correctly interpret *km*'s results, it is recommended to account for caveats inherent to RNA-Seq. For example, the presence of non-spliced transcripts, resulting in reads corresponding to intron sequences being reported by *km* as insertions in the *target sequence*. These sequences are easily detected by their specific genomic positions, lengths, and nucleotide sequences across different samples. We encountered this case during the FLT3–ITD analysis, where the *target sequence* overlaps exons 13 to 15 to cover all ITD locations (see the Target sequence in Materials and Methods section). Consequently, we have removed from *km*'s output 86-bp and 90-bp insertions, found at locations chr13:28,034,083 and chr13:28,034,304, respectively, corresponding to known introns (identified in 76 of 151 TCGA AML and 10 of 437 Leucegene samples).

Another scenario arises when the variant sequence is largely underrepresented compared with the *target sequence*. This can be due to the variant being present in a rare subclone or to strongly biased allele-specific expression. In this case, we can lower the coverage ratio threshold (default $P = 0.05$) to avoid ignoring these variants. The missing Leucegene NPM1 mutated sample (13H065) is such a case, where *km* was able to detect a tg/GTCCGA indel using higher sensitivity thresholds ($P = 0.01$). This variant has a local coverage ratio of 4% in this sample, just below the default threshold of 5%. Similar cases were identified for DNMT3A, where our *km* analysis of the TCGA cohort (Network et al, 2013) missed the identification of seven clinically annotated samples (of 18 with available RNA-Seq). By adjusting *km*'s parameters for higher sensitivity ($P = 0.01$ and $c = 2$), three of those samples were correctly identified.

### Detection of rearrangements

Detection of rearrangements bringing together two distant regions of the genome (such as translocations, large inversions, duplications, or deletions) requires a slightly different approach to the design of the *target sequence*. In the case of translocations (such as NUP98–NSD1), neither genomic nor transcript sequences can be used as a source for the *target sequence*. Instead, we created an artificial sequence that represents an expected transcript of the fusion. This artificial *target sequence* is then used to verify that each k-mer is covered in the sample's count table. With this approach, support for the *target sequence* is interpreted as a confirmation that this fusion is present in the sample. For large duplications or deletions, the same strategy is applied to avoid capturing several mutations (e.g., common single-base mutations) by using excessively large *target sequences*.

We applied this approach to the detection of the NUP98–NSD1 fusion and report the results in Table 2. We identified six samples in the Leucegene cohort that express a fusion transcript joining exon 11 of NUP98 to exon 7 of NSD1 (Lavallée et al, 2016). An alternative inserted sequence, which matches exactly exon 12 of NUP98, is also identified for seven samples (including the previous six). These results suggest alternative splicing events occurring in the fusion transcript. In the TCGA AML cohort, clinical analyses (Network et al, 2013) report an NSD1–NUP98 fusion in samples TCGA-AB-2930 and TCGA-AB-2856 and an NUP98–NSD1 fusion in TCGA-AB-2930. By adding a second *target sequence* to identify the fusion on the reverse strand, we identified all NSD1–NUP98 fusions but not the one on NUP98–NSD1. This can be explained by a fusion transcript undetectable with this type of *target sequence* (i.e., a fusion involving at least one exon at the extremities of the *target sequence*, e.g., exon 10 and 8 of NUP98 and NSD1, respectively) or a sequencing coverage too low to catch this variant.

The KMT2A–PTD is a tandem duplication that spans from exon 2 to exon 8 or 10 of KMT2A. Here, applying the strategy used for FLT3–ITD would require a *target sequence* of about 3,712 bp. Instead, we designed our *target sequence* by joining exon 8 directly to exon 2 to capture the specific junctions created by the duplication. Using this sequence, *km* identifies all versions of this duplication by returning an insertion if more exons are included in the duplicated region (Table 2). Surprisingly, further investigation of *km*'s output revealed that among the 24 samples identified as having a KMT2A–PTD in the Leucegene cohort (Lavallée et al, 2015), 6 samples showed evidence for more than one transcript. On the TCGA AML cohort, no report of this mutation has been found in cBioPortal or GDC (which combines clinical [Network et al, 2013] and computational analyses) but *km* identified three samples with a tandem duplication that includes up to exon 10.

# Discussion

To our knowledge, *km* is the first simple and unified method to detect variants arising from various sequence rearrangements (e.g., substitutions, duplications, inversions, and fusions) on RNA sequencing data. For this, *km* uses any sequence as an expected reference and reports all alternative forms sharing the reference's extremities. This approach is strictly restricted to the region analyzed but not to the type of variant to be identified. Moreover, this strategy improves the sensitivity for detecting challenging cases such as FLT3–ITD where position and length are highly variable.

The small amount of computational resources and time required by *km* to process and detect variants in 10,844 samples is unparalleled. Indeed, since the advent of whole genome and transcriptome sequencing, the default approach has been to perform analyses globally, resulting in very resource-intensive processes. Instead, *km* performs a deeper analysis on focused parts of the genome, preselected based on each user's interest (e.g., valuable for prognosis or targeted therapy). Thereby, *km* avoids consuming resources on a priori irrelevant parts of the genome and returns no results outside selected regions. Nonetheless, *km* analyses are incremental and allow testing of new regions and enrichment of

previous annotations for new variants. Unsurprisingly, our approach is sensitive to a lack of coverage over the target region, either from allelic imbalance or cellular heterogeneity. In these situations, the design of the RNA sequencing (e.g., depth and number of cells) needs to be adjusted for increased sensitivity.

Finally, this study shows *km*'s potential to establish fast and detailed diagnostics for any given patient, using only one sequencing, further extending the usefulness of these techniques in clinical settings. Indeed, all variants used to evaluate *km* have been shown to have prognostic values (Thiede et al, 2006; Bacher et al, 2008; Lavallée et al, 2016; Shiba et al, 2016; Yuan et al, 2016) for AML. As shown here, identifying these variants in 437 samples can successfully be done in 4 d with *km*, using a single workstation. Moreover, as sequencing becomes cheaper and more available, we envision a transition from targeted to whole transcriptome sequencing, followed by targeted analyses using methods like *km*. This will provide fast clinical prognostics but also gather the complete data related to a patient's disease for complementary analyses, to uncover other mutations during treatment, as well as for large-scale research. In the future, selecting and testing new target variants to extend the existing catalog would be an interesting step to extend *km*'s clinical (and research) applications to other diseases.

# Materials and Methods

### RNA sequencing of AML samples

This study is part of the Leucegene project, an initiative approved by the Research Ethics Boards of Université de Montréal and Maisonneuve-Rosemont Hospital. All AML samples were collected with informed consent between 2001 and 2015 according to Québec Leukemia Cell Bank procedures. RNA-Seq data were deposited in the Gene Expression Omnibus (GSE49642, GSE52656, GSE62190, and GSE67040) and exome sequencing in the Short Read Archive (BioProject PRJNA358716). Workflow for sequencing, mutation analysis, and transcript quantification has been described previously (Lavallée et al, 2015). Briefly, libraries were prepared with TruSeq RNA Sample Preparation kits (Illumina) and sequencing was performed using an Illumina HiSeq 2000 with 200-cycle paired-end reads. The complete dataset consists of 437 RNA-Seq samples (204 M reads per sample, on average).

TCGA data were obtained from the GDC portal for all 33 cancer types and preprocessed using Picard to obtain raw sequences from the available aligned files, which were then converted to Jellyfish count tables. The complete dataset represents 10,407 RNA-Seq samples, of which 151 correspond to AML (TCGA-AB-2975 sample was not in the initial TCGA AML cohort [Network et al, 2013]). TCGA results shown here are wholly or in part based upon data generated by the TCGA Research Network: http://cancergenome.nih.gov/.

### K-mer count tables with jellyfish

We used Jellyfish (v2.1.4) (Marcais & Kingsford, 2011) to create canonical k-mer count tables of length 31 bp, and opted not to store

**Life Science Alliance**

k-mers that were seen in a single read (parameters: –m 31 –C –L 2). These occurrences either result from sequencing errors or from very low coverage regions, and in both cases, cannot be leveraged to confirm the presence of a mutation. Moreover, sequencing errors are managed by asking Jellyfish to filter out k-mers containing at least one low-quality base (parameter: –Q+).

### *km*: k-mer walking as a local assembly

The main idea behind the implementation of a k-mer walking algorithm is to identify, in the raw sequencing reads, variant sequences that overlap with the two extremities of a *target sequence* designed to interrogate a region of interest. This *target sequence* can be broken down into k-mers overlapping by *k-1* bp to produce a linear directed graph where vertices represent k-mers and edges represent overlaps. Using this graph and a k-mer count table, diverging paths that connect the starting k-mer to the ending one can be identified. In the absence of an alternative path, the *target sequence* does not have a variant. Consequently, the *target sequence* needs to flank the variant by at least *k* bp on each side, to have this common starting and ending k-mer. Naturally, *k* must be large enough to linearly decompose the *target sequence*.

The k-mer walking algorithm is implemented as a depth-first search. To limit search space, we ignore branches that do not come back to a k-mer derived from the *target sequence* within *s* steps (default: 500). Also, new branches are explored only if each k-mer has a count greater than a fixed threshold *c* (default: 5) and greater than a fraction *P* (default: 5%) of the alternative k-mer count. These parameters may need to be adjusted to support detection of certain variants (e.g., as we did to find the indel on 13H065, see the Sensitivity and coverage in RNA-Seq section), but we have found that the default values perform well in general.

To extract variant sequences from the directed graph, we assign small weights (i.e., 0.01) to edges that are part of the *target sequence*, whereas the others are assigned a unit weight. This ensures the preferential use of paths from the reference sequence, and variants are enumerated by finding the set of unique shortest paths covering all edges of the graph.

The k-mer count table is the core data structure used by our approach and its content, construction time, and storage requirements represent key operational characteristics in using *km*. Once the k-mer count tables are prepared, the process of identifying variants is typically IO-bound until the queried count table is entirely cached in RAM. From an operational perspective, it becomes important to complete the analysis of all *target sequences* for each sample, before moving to the next sample. Alternatively, we have obtained great performance by copying count tables to a RAM disk (or by ensuring that it is cached using a simple scanning command such as "wc -l") before launching a series of targeted analyses. The speedup observed for using *km* is largely dependent on the number of regions to be investigated, but in absolute terms, preparation of the k-mer count table requires around 3 s per million reads of sequences and 0.1 s per samples for each target with length <300 bp (preloading of the jellyfish database to RAM requires around 25 s).

### Read dependency

An important caveat when identifying variants based on k-mer count tables is that we ignore read dependency between k-mers. During the k-mer walking, the overlap by k-1 bp between two k-mers is taken as an evidence that there exists at least one read that support the assembled sequence. In a De Bruijn graph (Compeau et al, 2011), the read dependency is explicitly stored in the graph, which explains the high memory requirements of this data structure. For *km*, the consequence of ignoring read dependency is that reducing *k* directly impacts the false-positive rate by increasing the frequency of incorrect reconstructions, which grows with the frequency of ambiguous regions of length *k-1*. The constraint that forces variant paths to begin and end on k-mers belonging to the *target sequence* acts as a very stringent filter to eliminate these spurious branches that would arise from low complexity regions of the genome. DiscoSNP (Uricaru et al, 2015) shares with *km* an internal step where variants are identified through k-mer–based local assembly. To provide a safety net, the authors of DiscoSNP opted to include a further step during which they align reads to variant sequences to compute k-read-coherency. Besides being a sound precaution, this read coherency will always remain limited to the read length, thus only protecting against incorrect identifications induced by ambiguous sequences of lengths between *k* and the read length (respectively 31 and 100 bp for the results presented here).

It can be very tempting to increase *k* (up to the read length) with the intention of increasing accuracy, but two factors need to be taken into consideration. First, any sequencing error throws away the contribution of *k* k-mers. Thus, as *k* is increased, the effective depth contributing to the analysis is decreased. Second, *km*'s algorithm cannot detect a variant that is located less than kbp away from the extremity of the *target sequence* because the variant branch would not merge back to the last k-mer. As a result, longer *target sequences* must be used, which can be problematic (see the case of NPM1, in the Target sequences design section). Although in a very different context (genome annotation from NGS), an optimal *k* was proposed to be between 19 and 20 to prepare k-mer count tables of the human genome. Using $k = 22$, a $10^{-4}$ probability of false location was also estimated (Philippe et al, 2009). These observations should be used as guiding principles in the rare cases, where *k* needs to be adjusted. Based on our experience in analyzing the Leucegene and TCGA cohorts, we have found few cases that prompted us to raise *k* from 21 to 31 as the project progressed. We have been so far satisfied with results obtained at $k = 31$. In a general case, we have no estimate of the frequency with which a variant would be wrongly identified at $k = 31$ and correctly disproved using the full reads. This subject remains to be further explored.

### Target sequences design

Creation of target sequences by an end user is relatively simple. In general, *k* bp from each side of the targeted variant are required for exact mutation identification, and larger regions may be used for more exploratory analyses. Our *target sequence* for the DNMT3A R882 mutation serves as a simple example in which 65 bp ($31 + 3 + 31 = 65$) represent the minimal *target sequence* covering the R882 codon.

The FLT3–ITD and the 4-bp insertion in NPM1 present more challenging situations. For NPM1, the expected mutation appears close to the end of the last exon, and we recommend extending the sequence as it is essential that the last k-mer of the *target sequence* does not overlap the targeted mutation (see the k-mer walking as a local assembly section). Consequently, we prepared an 80-bp *target sequence* comprising the end of exon 10, exon 11, and 14 bp of the 3′-UTR. For FLT3–ITD, the duplication can span over two exons (14 and 15) and can even be longer than sequencing reads. To cover all insertion points and lengths, we prepared a 345-bp *target sequence* overlapping exons 13 to 15 of the FLT3 transcript. For detection of rearrangements, we need to include kbp on each side of the junction created by the targeted variant (see the Detection of rearrangements section).

To assess the fraction of the transcriptome that can be targeted using *km*, we computed the fraction of unique sequences identified for various values of $k$ between 10 and 100. With $k = 31$, 96.76% of the human transcriptome is readily represented by a linear k-mer graph and thus accessible as a *target sequence* for *km* (see Fig S2).

## Supplementary Information

## Acknowledgements

The authors wish to thank Muriel Draoui for project coordination and Sophie Corneau for sample coordination, Isabel Boivin for data validation as well as Marianne Arteau and Raphaëlle Lambert at the Institute for Research in Immunology and Cancer genomics platform for RNA sequencing. The dedicated work of BCLQ staff, namely, Giovanni d'Angelo, Claude Rondeau, and Sylvie Lavallée is also acknowledged. This work was mostly supported by Genome Canada and Genome Quebec with supplementary funds from Amorchem. Contribution from Ministère de l'économie, de l'innovation et des exportations du Québec, and Leukemia Lymphoma Society of Canada is acknowledged. G Sauvageau and J Hébert are recipients of research chairs from the Canada Research Chair program and Industrielle-Alliance (Université de Montréal). BCLQ is supported by grants from the Cancer Research Network of the Fonds de Recherche du Québec – Santé. RNA-Seq read mapping and transcript quantification were performed on the supercomputer Briaree from Université de Montréal, managed by Calcul Québec and Compute Canada. The operation of this supercomputer is funded by the Canada Foundation for Innovation, NanoQuébec, Réseau de Médecine Génétique Appliquée, and the Fonds de Recherche du Québec - Nature et Technologies. V-P Lavallée is supported by a fellowship from the Cole Foundation and by a Vanier Canada Graduate Scholarship.

### Author Contributions

EO Audemard: software, methodology, and writing—original draft, review, and editing.
P Gendron: resources, software, methodology, and writing—review and editing.
A Feghaly: software and writing—review and editing.
V-P Lavallée: resources, methodology, and writing—review and editing.
J Hébert: resources, funding acquisition, and validation.
G Sauvageau: conceptualization, funding acquisition, and writing—review and editing.
S Lemieux: conceptualization, software, funding acquisition, methodology, and writing—original draft.

### Conflict of Interest Statement

The authors declare that they have no conflict of interest.

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
