## [Reviewer comments · Life Science Alliance]

Life Science Alliance

Targeted variant detection using unaligned RNA-Seq reads

Éric Audemard, Patrick Gendron, Albert Feghaly, Vincent-Philippe Lavallée, Josée Hébert, Guy Sauvageau, and Sébastien Lemieux

DOI: [10.26508/lsa.201900336](https://doi.org/10.26508/lsa.201900336)

Corresponding author(s): Sébastien Lemieux, University of Montreal

Review Timeline:	Submission Date:	2019-02-08
	Editorial Decision:	2019-04-11
	Revision Received:	2019-07-17
	Editorial Decision:	2019-07-29
	Revision Received:	2019-08-01
	Accepted:	2019-08-02

Scientific Editor: Andrea Leibfried

Transaction Report:

April 11, 2019

Re: Life Science Alliance manuscript #LSA-2019-00336-T

Prof Sébastien Lemieux
University of Montreal
IRIC
P.O. Box 6128 Station Centre-Ville
Montreal, QC H3C 3J7
Canada

Dear Dr. Lemieux,

Thank you for submitting your manuscript entitled "Targeted variant detection using unaligned RNA-Seq reads" to Life Science Alliance. The manuscript was assessed by expert reviewers, whose comments are appended to this letter. I apologize that the process took so long, it took a while to secure reviewers in this case. I am happy to say that all three reviewers provided very constructive input, based on which we would like to invite you to submit a revised version of your manuscript.

Importantly, a revised version should satisfactorily:

- benchmark/compare the method extensively and in a valid way, demonstrating superiority
- show that the method can be easily applied (easier installation is needed as well as more information on how to use it, including necessity to perform controls)
- demonstrate the broad applicability of the method

Thank you for this interesting contribution to Life Science Alliance. We are looking forward to receiving your revised manuscript.

Sincerely,

B. MANUSCRIPT ORGANIZATION AND FORMATTING:

Reviewer #1 (Comments to the Authors (Required)):

In "Targeted variant detection using unaligned RNA-Seq reads", the

authors present an analysis of mutations in Acute Myeloid Leukemia (AML) and the tool, km, they developed for that purpose. The km tool is designed to be of general interest as an alignment-free method to detect variations in RNA-Seq data.

The speed and accuracy of the km tool is obtained by directly focusing on "target sequences": sequences extracted or synthesized from the transcriptome containing only the region of interest.

Although the km tool could be of practical interest, it is unclear if the benefits are coming from the local assembly procedure rather than the appropriate design of target sequences.

Major Comments

=====

* The main claim for the km software is to be faster and more accurate thanks to focusing on targeted sequences. This is in contrast to the traditional method of aligning the sequences reads to the entire genome or transcriptome, then calling the variances.

Given that the target sequences of interest are known, the fair comparison is not by aligning to the whole genome, but to the target sequences directly. Does km performs better (faster? less memory usage? more accurate?) than an existing pipeline where the alignment is done against the target sequences.

In other words, comparing a "de novo" variance detection by looking at the entire genome with a targeted variance detection where the target sequences are known, is not a valid comparison.

* The method of local assembly, as is done by km, has been used previously. For example, software like the ALLPATHS genome assembler consider all the possible paths between two localized anchors. The SuperReads method in the MaSuRCA assembler fills in mate pair by looking at all the possible paths between the mate pairs of an insert. Extra verification on the validity of a paths are performed, for example by looking at the forward and reverse paths found.

There should be more comparison and contrast with existing methods for local assembly.

Minor Comments

=====

* Page 3: "to produce a linear directed graph". Add a word saying that the k-mer size is chosen so that the graph is linear.

* Page 5: "In contrast, aligning reads for all samples using STAR...". This ties to the first major comment above. It needs to

clearer here that the alignment is done against the entire genome, not just the target sequences.

* Table 2: the meaning of the thin horizontal line separating the last three rows should be added to the caption (not just in the main text).

* Table 2, Leucegene, last 3 rows: The numbers in the Variant and Target column do not make sense, and the main text explanation on page 6 on how to interpret them is not sufficient.

How can Variant be larger than Target? If a variant is found, doesn't it mean that the sample expresses the target sequence (or a variation of it), as is explained in the caption?

Reviewer #2 (Comments to the Authors (Required)):

The manuscript by Audemard et al. describes a high throughput bioinformatics software tool that allows for the rapid identification of single-nucleotide variants (SNV), small insertions and deletions (INDELs), duplications, and rearrangements from non-aligned raw data sets (FASTA files). We find the tool developed by Audemard et al., to be useful for situations in which a researcher wishes to rapidly query large sets of raw data for specific mutations.

We felt that the software was only moderately easy to install. On a system with root access it was easy and could be completed in a few minutes. However, on an institutional server (without root access) downloading the major requirement (jellyfish) and subsequently applying the required python bindings - was not straightforward and could pose as an obstacle to more widespread use.

Concerns:

1) Documentation of the program is one concern. The manuscript and github pages are devoid of any information pertaining to the generation of "target" sites. Based on the example (run_leucegene.sh), it seems the requirement for the "catalog" is FASTA files covering the targets of interest. In some cases, the example files include two different regions (perhaps due to splicing of exons?) Generating functional catalog files is a necessary aspect of running the km program and not including any information as to how exactly generating these target catalogs is done is worrisome. Furthermore, it is unclear whether there are any specific requirements for this step? How long should the FASTA sequence covering a mutation be? Is there a maximum length?

2) A test suite that involves downloading a 46 GB file is also somewhat unreasonable/unnecessary... Perhaps downloading a smaller test dataset is possible?

3) We ran the pipeline on our own dataset (75 bp single-end RNA-seq reads) verifying a highly prevalent (50% allele frequency) SRSF2 (P95H) mutation. Using a k-mer length of 31 (default) or 25 during the jellyfish count step resulted in the program COMPLETELY missing our mutation and incorrectly reporting only the presence of reference sequence. After setting the k-mer size to 20 the program finally returned accurate results. Perhaps more explanation, in the manual, on how to optimally set program parameters would be beneficial.

Other notes:

-g parameter for find_mutation command requires a Python library that is not explicitly installed (matplotlib.pyplot) or listed as required.

Reviewer #3 (Comments to the Authors (Required)):

The manuscript describes a method for rapid detection of mutations and variants using NGS data in a short, targeted, region. Overall, the data presented by the authors demonstrate that their method has a potential for accelerating targeted mutation analysis. The drawbacks of the general approach are obvious: The vast majority of information that sequencing of a tumor provides is disregarded, only small genomic region is analyzed, and therefore, it may lead to tunnel vision. The biggest weakness of the method, as I understand it, is that complex rearrangements and insertions require the preparation of a custom target sequence. Those will be available only to prevalent insertions, which most likely represent only a fraction of the patient samples (i.e., most cancers do not have the same underlying mutation). However, these are limitations of the approach by design and the authors acknowledge, therefore potential users would be able to make an informed choice. Even if the method has only limited use, there is still value in it, especially for detection of small genomic changes.

Major comments

The km algorithm that is presented very briefly in the results section, and authors almost immediately present statistics about km runtime. It is extremely important that in the results there would be a clear outline of the algorithm with few major use cases. This would require expanding Figure 1, as currently it does a poor job in illustrating how would rearrangements, fusions, and large insertions are detected.

The authors did not show convincing data that their approach could detect large insertions, and it is unclear how would their approach handle insertions larger than the length of the kmer.

Similar to #2, in case of rearrangement/fusion, the preparation of a target sequence that includes the fusion, is very limiting potential use cases.

Minor comments

The authors point out that the presence of unprocessed mRNAs may contaminate the analysis with intron sequences, which could be interpreted as insertions. Instead of having users to manually identify this as a potential problem, the authors are encouraged to add an automated filtering step based on intron annotation of the target species. The kmer based approach should work well for that as well.

Considering the former point, how would km handle mutations that result in intron inclusion in the mature sequence?

Language appeared a bit exaggerated and requires editing. Examples:

Introduction, first paragraph: notoriously difficult

Introduction. "This mapping step leads to frequent mapping errors".. I would replace "leads to" with "may introduce".

Reviewer #1

The main claim for the km software is to be faster and more accurate thanks to focusing on targeted sequences. This is in contrast to the traditional method of aligning the sequences reads to the entire genome or transcriptome, then calling the variances. Given that the target sequences of interest are known, the fair comparison is not by aligning to the whole genome, but to the target sequences directly. Does km performs better (faster? less memory usage? more accurate?) than an existing pipeline where the alignment is done against the target sequences.

- We agree with the reviewer that the comparison was unfair as presented and have removed it from the paper. Instead, we opted to directly report resources required by km to analyse 437 samples (page 6, first paragraph). Our goal here is to highlight the "practicality" of this type of analysis on relatively large cohorts.
- Out of curiosity, we implemented the suggestion by reviewer #1 to apply a mapping-based pipeline (STAR 2.6.1d) in which the reference would only contain the target sequences. Running times (for mapping only) were not reduced compared to a full genome application (memory usage was much lower). This result is expected as STAR (and others) uses a data structure (suffix array) that has a search complexity independent of the genome size. We felt that this result was beside the scope of the paper and opted to leave it out.

The method of local assembly, as is done by km, has been used previously. (...) There should be more comparison and contrast with existing methods for local assembly.

- Thanks for pointing this out, we have modified the manuscript to insert appropriate references highlighting similarities between km and local assembly variant calling methods (last paragraph of Introduction, page 3). Appropriate citations were also added where we mention local assembly algorithms (first paragraph in "Overview of km and target sequence", page 3).

Page 3: "to produce a linear directed graph". Add a word saying that the k-mer size is chosen so that the graph is linear.

- We modified the text accordingly: "... **with k large enough** to produce a linear directed graph" (page 3).

Page 5: "In contrast, aligning reads for all samples using STAR...". This ties to the first major comment above. It needs to be clearer here that the alignment is done against the entire genome, not just the target sequences.

- This part has been removed, see our response above.

Table 2: the meaning of the thin horizontal line separating the last three rows should be added to the caption (not just in the main text).

- Thanks for pointing this out, we have modified the table and caption accordingly. We took this opportunity to make sure that the difference in the types of targets better stands out throughout the manuscript. For this, we introduced an explicit naming scheme: "reference targets" and "variant targets", and adjusted the main text accordingly.

Table 2, Leucegene, last 3 rows: The numbers in the Variant and Target column do not make sense, and the main text explanation on page 6 on how to interpret them is not sufficient. How can Variant be larger than Target? If a variant is found, doesn't it mean that the sample expresses the target sequence (or a variation of it), as is explained in the caption?

- To clarify interpretation of these numbers, we have made several modifications to both figures, captions and main text (mainly in Fig. 1, also addressing a comment from Reviewer #3). Briefly, if the sample has a homozygous variant, it will be counted only in variant column as the target sequence is not expressed.

Reviewer #2

We felt that the software was only moderately easy to install. On a system with root access it was easy and could be completed in a few minutes. However, on an institutional server (without root access) downloading the major requirement (jellyfish) and subsequently applying the required python bindings was not straightforward and could pose as an obstacle to more widespread use.

- We'd like to thank reviewer #2 for this feedback on the installation process. We have updated the documentation and added an easy install section and script to simplify the process of installing km without root privileges.

1) *Documentation of the program is one concern. The manuscript and github pages are devoid of any information pertaining to the generation of "target" sites. Based on the example (run_leucegene.sh), it seems the requirement for the "catalog" is FASTA files covering the targets of interest. In some cases, the example files include two different regions (perhaps due to splicing of exons?) Generating functional catalog files is a necessary aspect of running the km program and not including any information as to how exactly generating these target catalogs is done is worrisome. Furthermore, it is unclear whether there are any specific requirements for this step? How long should the FASTA sequence covering a mutation be? Is there a maximum length?*

- We have substantially updated the documentation by specifically adding a section detailing the strategies for designing target sequences (See "Design your target sequence").
- For further assistance, we have also implemented a web application for preparing human and mouse target sequences ([km-target](https://bioinfo.irc.ca/km-target/) <https://bioinfo.irc.ca/km-target/>).

2) *A test suite that involves downloading a 46 GB file is also somewhat unreasonable/unnecessary... Perhaps downloading a smaller test dataset is possible?*

- We add a test section to the documentation which runs on a very small datasets that is now included in the github repository along the source code. These datasets are pre-computed jellyfish tables prepared from subsets of reads mapping to the region of interest. They should only be used for testing purpose.

3) *We ran the pipeline on our own dataset (75 bp single-end RNA-seq reads) verifying a highly prevalent (50% allele frequency) SRSF2 (P95H) mutation. Using a k-mer length of 31 (default) or 25 during the jellyfish count step resulted in the program COMPLETELY missing our mutation and incorrectly reporting only the presence of reference sequence. After setting the k-mer size to 20 the program finally returned accurate results. Perhaps more explanation, in the manual, on how to optimally set program parameters would be beneficial.*

- We are sorry for giving you so much trouble. As mentioned above, the documentation has been significantly expanded and should resolve this situation. We also created a web application to prepare human and mouse target sequences: [km-target](https://bioinfo.irc.ca/km-target/) (<https://bioinfo.irc.ca/km-target/>). In the eventuality that these

supplementary resources are not enough, we would like to encourage the reviewer to contact us or open an issue on github. We have tried to be very pro-active in helping users to fine-tune target sequences for various variants.

-g parameter for find_mutation command requires a Python library that is not explicitly installed (matplotlib.pyplot) or listed as required.

- This option has been removed, it was a prototype and was not meant to be available in the release version of km as we found that the interpretation of the resulting graphic tends to confuse more than inform.

Reviewer #3

The km algorithm that is presented very briefly in the results section, and authors almost immediately present statistics about km runtime. It is extremely important that in the results there would be a clear outline of the algorithm with few major use cases. This would require expanding Figure 1, as currently it does a poor job in illustrating how would rearrangements, fusions, and large insertions are detected.

- As suggested by reviewer #3, we have significantly expanded Figure 1 (page 4) and the text of section “Overview of km and target sequences” (pages 3-5). We believe this new version should clearly outline how rearrangements and large insertions are detected.

The authors did not show convincing data that their approach could detect large insertions, and it is unclear how would their approach handle insertions larger than the length of the kmer.

- We have modified Figure 1 to better explain the approach used to detect insertions, a process that is limited neither by read or k-mer lengths. Supplementary table S3 demonstrates this by reporting a majority of FTL3-ITDs (essentially insertions of variable lengths) of lengths greater than 50 bp (read length for this dataset, TCGA-AML, was 50 bp). This, consequently, demonstrates that insertions longer than 31 bp (k-mer length used) can be identified.

The authors point out that the presence of unprocessed mRNAs may contaminate the analysis with intron sequences, which could be interpreted as insertions. Instead of having users to manually identify this as a potential problem, the authors are encouraged to add an

automated filtering step based on intron annotation of the target species. The kmer based approach should work well for that as well.

- Although we acknowledge the appeal of this suggestion, it's implementation would unfortunately not be trivial and could risk limiting km's flexibility. For example, in several contexts, intron retention or discovery of an alternate splice site are of interest. Automation in the interpretation of variants is on top of our "to-do" list but we felt that an initial version should favor simplicity and flexibility.
- As we find proper ways to introduce this type of utilities, we will add them to the github.

Considering the former point, how would km handle mutations that result in intron inclusion in the mature sequence?

- Here, two approaches could be used, depending on the level of information desired. First would be to use the spliced form as a query. The splice-inhibiting variant mentioned would first have higher coverage of the intron sequence and the insertion would contain the specific mutation. If the identification of this specific mutation is of interest, then the query should be the unspliced variant.

Language appeared a bit exaggerated and requires editing. Examples: notoriously difficult, This mapping step leads to frequent mapping errors" (...).

- We agree with reviewer #3 and have revised our text, particularly the introduction, to avoid these overstatements.

July 29, 2019

RE: Life Science Alliance Manuscript #LSA-2019-00336-TR

Dr. Sébastien Lemieux
University of Montreal
Institute for Research in Immunology and Cancer (IRIC)
P.O. Box 6128, Centre-Ville
Montreal, QC H3C 3J7
Canada

Dear Dr. Lemieux,

Thank you for submitting your revised manuscript entitled "Targeted variant detection using unaligned RNA-Seq reads". As you will see, reviewer #2 is happy with the revision performed and we would thus be happy to publish your paper in Life Science Alliance pending final revisions necessary to meet our formatting guidelines.

- please upload your manuscript text as a word docx file, the tables can remain in this file
- please upload the figures, including suppl figures, as individual files and without figure legends; figure legends should remain in the word docx file

A. FINAL FILES:

-- Summary blurb (enter in submission system): A short text summarizing in a single sentence the study (max. 200 characters including spaces). This text is used in conjunction with the titles of papers, hence should be informative and complementary to the title. It should describe the context

and significance of the findings for a general readership; it should be written in the present tense and refer to the work in the third person. Author names should not be mentioned.

B. MANUSCRIPT ORGANIZATION AND FORMATTING:

Sincerely,

Andrea Leibfried, PhD
Executive Editor
Life Science Alliance
Meyerohofstr. 1
69117 Heidelberg, Germany
t +49 6221 8891 502
e a.leibfried@life-science-alliance.org
www.life-science-alliance.org

Reviewer #2 (Comments to the Authors (Required)):

We appreciate the authors' substantial efforts in providing additional documentation to accompany their software tool, simplifying the installation methods, shrinking the test dataset, and further addressing the concerns of the other reviewers. We think this tool could be used by other members of the scientific community and thus would be benefit in being published.

August 2, 2019

RE: Life Science Alliance Manuscript #LSA-2019-00336-TRR

Dr. Sébastien Lemieux
University of Montreal
Institute for Research in Immunology and Cancer (IRIC)
P.O. Box 6128, Centre-Ville
Montreal, QC H3C 3J7
Canada

Dear Dr. Lemieux,

Thank you for submitting your Methods entitled "Targeted variant detection using unaligned RNA-Seq reads". It is a pleasure to let you know that your manuscript is now accepted for publication in Life Science Alliance. Congratulations on this interesting work.

DISTRIBUTION OF MATERIALS:

Again, congratulations on a very nice paper. I hope you found the review process to be constructive and are pleased with how the manuscript was handled editorially. We look forward to future exciting submissions from your lab.

Sincerely,
